# Progress in Prophylactic and Therapeutic EBV Vaccine Development Based on Molecular Characteristics of EBV Target Antigens

**DOI:** 10.3390/pathogens11080864

**Published:** 2022-07-30

**Authors:** Marija Rozman, Petra Korać, Karlo Jambrosic, Snjezana Židovec Lepej

**Affiliations:** 1Department of Immunological and Molecular Diagnostics, University Hospital for Infectious Diseases Zagreb, Zagreb 10000, Croatia; marijarozman114@gmail.com; 2Division of Biology, Faculty of Science, University of Zagreb, Zagreb 10000, Croatia; petra.korac@biol.pmf.hr; 3Laboratory for Analytical Chemistry and Biogeochemistry of Organic Compounds, Division for Marine and Environmental Research, Ruđer Bošković Institute, Zagreb 10000, Croatia; Kjambros@irb.hr

**Keywords:** EBV replication cycle, EBV and immune response, prophylactic EBV vaccine, therapeutic EBV vaccine, EBV mRNA vaccine

## Abstract

Epstein–Barr virus (EBV) was discovered in 1964 in the cell line of Burkitt lymphoma and became first known human oncogenic virus. EBV belongs to the *Herpesviridae* family, and is present worldwide as it infects 95% of people. Infection with EBV usually happens during childhood when it remains asymptomatic; however, in adults, it can cause an acute infection known as infectious mononucleosis. In addition, EBV can cause wide range of tumors with origins in B lymphocytes, T lymphocytes, and NK cells. Its oncogenicity and wide distribution indicated the need for vaccine development. Research on mice and cultured cells as well as human clinical trials have been in progress for a few decades for both prophylactic and therapeutic EBV vaccines. The main targets of the vaccines are EBV envelope glycoproteins such as gp350 and EBV latent genes. The long wait for the EBV vaccine is due to the complexity of the EBV replication cycle and the wide range of its host cells. Although some strategies such as the use of dendritic cells and recombinant Vaccinia viral vectors have shown success, ongoing clinical trials using mRNA-based vaccines as well as new delivery systems as nanoparticles are yet to show the best choice of vaccine target and its production strategy.

## 1. Introduction

Epstein–Barr virus (EBV), also known as *Human gammaherpesvirus 4,* is a ubiquitous virus from the *Herpesviridae* family and *Lymphocryptovirus* genus. It is the first identified oncovirus with high cell-transforming ability discovered in cultured Burkitt lymphoma tumor cells [1] Moreover, it is the first virus from the *Herpesviridae* family with cloned and sequenced genome, and today is included as a group 1 carcinogen [2]. EBV infects around 95% of people worldwide, most of them in childhood, when it usually remains asymptomatic. Among young adults without previous EBV contact, it can cause an acute infection represented as infectious mononucleosis, characterized by lymphadenopathy, fever, and swollen liver or spleen. Once infected, an individual remains positive for EBV for the rest of their life, with possible reactivation of EBV due to the weakness of the immune system. In addition, subsequent activation from EBV latency can lead to malignancies such as Hodgkin’s lymphoma, nasopharyngeal cancer, lymphoproliferative diseases, gastric carcinoma, and endemic and sporadic Burkitt lymphoma, as well as a wide range of lymphomas that have origin in T lymphocytes or NK cells. EBV also causes post-transplant lymphoproliferative disorders (PTLDs), with all reported PTLD cases being EBV-positive [3]. Its oncogenic potential is a result of precisely regulated latencies established according to its microenvironment and immune status of the host. In addition, EBV infection is linked to a higher risk of multiple sclerosis development, confirmed by longitudinal analysis over a 20-year period [4], as well as autoimmune diseases. Moreover, the recent and still ongoing COVID-19 epidemic a provided study showing a positive correlation between EBV viremia and memory-related incidence of long COVID-19 symptoms [5], as well as more severe pneumonia among patients with severe COVID-19 infections who were EBV-positive in comparison with EBV-negative patients [6]. Among virologists, the EBV vaccine has already for a few decades been a subject of research; however, a licensed vaccine is still not in use.

### 1.1. EBV Structure and Tropism

EBV has 172 kb long double-stranded DNA with approximately 500 bp long terminal repeats that allow its circularization, leading to episome formation after entering the host cell [7]. EBV genome has variability of around 0.002%, which makes it stable. Most mutations are found in latency genes linked to different strains and geographical regions, as well as malignancy risk. Its genome is protected by the nucleocapsid, and it additionally contains a lipid bilayer envelope and tegument (Figure 1). The tegument is a set of proteins between the nucleocapsid and envelope that have a significant role in its replication cycle [8]. Protein BGLF2, positioned in the tegument, promotes EBV reactivation [9], while BNRF1 tegument protein is important in viral transport from the endosome into the nucleus of B lymphocytes [10]. The envelope contains several types of glycoproteins that are important in recognition and fusion with B lymphocytes such as gp350, gH, gL, gB, and gp42 (Figure 1). B lymphocytes are the main target of EBV; however, the virus usually replicates in the epithelial cells of the oropharynx before infecting B lymphocytes [11]. During entry into epithelial cells, EBV fuses its membrane directly to the host cell membrane, where gB activates fusion while the complex gH/gL regulates it [12]. This process is different from entry into B lymphocytes, where more glycoproteins participate in fusion and the mechanism is better understood. Additionally, epinephrin receptor A2 showed significance in epithelial cell infection as its overexpression increased infection and its knockdown reduced it [13]; however, this receptor showed significance in the reproducibility of infection only in cancer-derived organoids [14].

Gp350 or gp350/220 is glycoprotein made in two forms, gp350 and gp220, respectively, as a result of mRNA alternative splicing. Gp350 is a highly glycosylated protein that is 901 amino acids long, with half of its mass being carbohydrate. Gp350 makes an initial connection to B lymphocytes by binding to their complement receptor type 2 (CR2) or CD21, respectively [15]. CR2 is a transmembrane protein found on B lymphocytes, T lymphocytes, and follicular dendritic cells; moreover, it plays a role in the development of humoral immunity response [16]. EBV main fusion proteins are gB, gL, and gH, which are found on all Herpesviruses. On the other hand, gp42 determines EBV tropism [5]. Gp42 binds major histocompatibility complex class II (MHC II) expressed on B lymphocytes, which triggers the fusion of membranes. Virions lose gp42 after they are exported from B lymphocytes and, therefore, are not able to reinfect another cell [17]. Those proteins are the main vaccine targets, as they are necessary for virus entry and, therefore, its primary infection [18]. Besides envelope glycoproteins and CR2, successful infection of B lymphocytes requires two additional cellular proteins: HLA and integrin [19]. HLA class II proteins on the host cell bind to the gHgL/gp42 complex, which activates membrane fusion during primary infection, while integrin triggers entry of EBV into epithelial cells with binding to gHgL [8]. EBV infection of epithelial cells also requires BMRF-2 protein on the EBV envelope, which binds to the β1 integrin family on the surface of polarized oral epithelial cells [20].

### 1.2. EBV Replication Cycle

The EBV replication cycle consists of the lytic and latent phase, precisely regulated according to the microenvironment and immunological status of an individual. The lytic phase is characterized by the production of virions, as well as the lysis of the infected cell, and is further divided into immediate-early, early, and late phases according to the different gene expression profiles. Protein Rta coded by *BRLF1* gene and Zta protein coded by *BZLF1* gene are expressed in the immediate-early phase as they are transcription factors for early genes. Besides its role in lytic cycle induction, protein Rta also stabilizes some proteins in capsid and if found in the tegument. In addition, Rta and Zta are key proteins for reactivation from the latent to lytic phase. Early genes code for enzymes necessary for DNA replication such as DNA polymerase and enzymes involved in nucleotide metabolism such as thymidine kinase *BXFL1*, but also for proteins involved in the expression of late genes. Late genes code for proteins needed for virion assembly, that is, viral antigens, protease, and glycoproteins, which are the components or their envelope [17]. During this phase, EBV forms concatemeric repeats, which are fragmented between terminal repeats before being encapsidated during virion formation [21]. On the other hand, the latency phase is characterized by the EBV dormant state with a small number of genes expressed and, according to the gene expression profile, divided in latency of 0, I, II, and III (Table 1). EBV inside initially infected B lymphocytes goes through latency type III, with the highest number of genes expressed compared with the other latency stages. In latency III, there is a co-expression of EBV nuclear antigens *EBNA1*, *EBNA2*, *EBN3A*, *EBNA3B*, and *EBN3C*, as well as latent membrane proteins *LMP1*, *LMP2A*, and *LMP2B* [22]. EBNA1 is necessary for replication, segregation, and persistence of episome inside the host cell, and it also plays a role in EBV-related oncogenesis. EBNA3 proteins are all coded with the same gene, but spliced after transcription; they play role in the activation and immortalization of B lymphocytes [23]. EBNA2 stimulates B lymphocyte proliferation and, with its activating domain, recruits transcription factors near the promoter site. EBNA3 proteins are all coded with the same gene, but spliced after transcription; they play a role in the activation and immortalization of B lymphocytes. LMPs induce signal transduction pathways that result in further proliferation of infected B lymphocyte and apoptosis avoidance. Latency type III is typical for cells with a high proliferative index, such as cells affected by lymphoproliferative diseases. Infected B lymphocytes go through latency type II inside germinal centers in secondary lymphatic tissues where *EBNA1*, *LMP1*, *LMP2A,* and *LMP2B* are expressed. After maturation in germinal centers, B lymphocytes can either synthesize plasma cells, which directs them to lytic reactivation, or, as an alternative, cells can go to latency 0, where only EBER mRNAs and no proteins are expressed [24]. EBERs include EBER1 and EBER2, which are small, non-coding, and non-polyadenylated RNAs that interact with certain cellular protein, which prevents apoptosis and plays a significant role in a wide range of EBV-related malignancies [25]. In addition, EBERs induce cytokines and modulate innate immune response. Latency 0 is important for infected B lymphocytes circulating inside the host peripheral blood because, with no proteins expressed, it cannot be recognized by the immune system. During the division of memory B lymphocytes, EBV enters latency I, where, besides EBERs, EBNA1 is expressed. Protein synthesis during certain latency types is maintained by Poly [ ADP-ribose] polymerase 1, which binds the EBV genome [26].

### 1.3. Interaction of EBV and the Immune System

Initial EBV infection usually happens in childhood when seroconversion happens, hence the production of neutralizing EBV antibodies [27]. Once inside the host, EBV begins interplay with the host immune system, switching between its latent and lytic state [28]. The strongest immune reaction to EBV is in the later stages of infection, during which certain autoimmune diseases, such as multiple sclerosis and type I diabetes, can arise. EBV initially infects epithelial cells inside the oro-pharynx, and subsequently the B lymphocytes inside, where it establishes a dormant state. Initial proliferation of B lymphocytes is a consequence of EBV’s precisely regulated latency genes [29]. Besides envelope glycoproteins, latent membrane protein 1 (LMP1) and latent membrane protein 2A (LMP2A) have a significant role in EBV infection. LMP1 mimics CD40 molecule on follicular CD4+ T lymphocytes, which causes the activation of signal transduction pathway, which is essential for B lymphocytes’ differentiation in plasma and memory cells. Additionally, LMP2A mimics signaling initiated by the B lymphocyte receptor, which permits their proliferation [30]. This eliminates the need for their activation with CD4+ lymphocytes. When EBV is in its lytic phase, B lymphocytes produce infectious virus particles. B lymphocytes present viral peptides on their HLA class I molecules, which CD8+ lymphocytes recognize, and they consequently destroy the infected cells. Moreover, B lymphocytes produce IgG/IgA antibodies against gp350/220, which prevents EBV attachment to CR2 [19]. The latent state is the stage of EBV replication cycle when the virus is the most difficult to detect by the immune system owing to the reduced number of proteins expressed.

## 2. EBV Vaccine Development

The correlation between EBV infection and the incidence of a wide range of lymphomas as well as infectious mononucleosis indicated the need for EBV vaccine development. However, until now, there is still no licensed vaccine in use. The complexity of the EBV replication cycle as well as its high number of envelope proteins make vaccine development a demanding task. EBV can infect T lymphocytes as well as NK cells through still unknown mechanisms, which makes the development of a vaccine that would prevent the infection of all EBV-target cells currently not possible [19]. In addition, EBV lacks a true animal model as its infection was observed only among humans. Cotton-top tamarins and marmosets were used as early animal models for EBV infection because their genes show high homology to those of human, but, owing to their endangerment, their use is no longer permitted. Research in 2017 showed the first signs of a potential animal model for EBV after 8 out of 10 tree shrews, mammals native to tropical forests, showed signs of infection after intravenous injection with EBV [31]. Accordingly, follow-up research showed that tree shrews have one hundred percent identity with humans in CR2 residues critical for attachment to gp350 [32]. Besides homology in CR2, tree shrews are suitable animal models because of their small size and short reproductive cycle as well as life span [33]. *BLLF1* codes for gp350 protein, necessary for attachment to B lymphocytes, which makes them a potential animal model for studying the course of EBV infection and its vaccine targets. There are two types of EBV vaccines going through clinical trials, a prophylactic vaccine and therapeutic vaccine. The aim of prophylactic vaccines is to prevent primary infection and induce adaptive immunity. The prophylactic vaccine would significantly reduce the spread of EBV to other hosts as its fusion with B lymphocytes would be eliminated. The main targets for prophylactic vaccine development are envelope glycoproteins because they are responsible for the initial attachment and fusion with B lymphocytes. However, an obstacle in prophylactic vaccine development is the lack of a control group, as only a minority of people have never been in contact with EBV. On the other hand, the aim of therapeutic vaccines is to stimulate immunological response to destroy EBV-infected cells. The therapeutic vaccine would reduce the development of infectious mononucleosis and EBV-related malignancies. However, the development of the prophylactic vaccine, by eliminating primary infection, would reduce EBV-related tumors as well. The main targets for therapeutic vaccine development are proteins expressed in the latent phase as well as proteins such as ZEBRA protein that are responsible for initiation of the lytic cycle and, therefore, EBV reactivation. Some vaccines contain an adjuvant, substances that increase the response of the immune system to target antigens, with the most common one being aluminum hydroxide, sometimes combined with monophosphoryl lipid A. Freund’s adjuvant and immune-stimulating complexes have also been used in order to enhance the defense against primary infection and later reactivation from the latent state. The choice of adjuvant has been shown to lead to significant differences in vaccine production [19].

An EBV vaccine is highly needed, and the need grows with recent research that shows its connections to other diseases such as multiple sclerosis and COVID-19. Autoimmune diseases such as acquired immunodeficiency syndrome (AIDS) make the host more susceptible to malignant transformations caused by EBV [34]. Other infections, such as malaria caused by *Plasmodium falciparum*, cause endemic Burkitt lymphoma if the host is additionally infected with EBV [35]. A wide range of host cells and its avoidance of the immune system make EBV a ubiquitous virus with a strong ability to survive and adapt its microenvironment to its own needs. Although most people will acquire EBV and remain asymptomatic, the presence of a wide range of malignancies caused by EBV indicates the need for the vaccine. EBV can weaken the immune system or make its host cells susceptible to malignant transformation by some other agent. In addition, EBV-related malignancies keep growing; in the period from 1990 to 2017, the incidence of gastrointestinal carcinoma, Hodgkin lymphoma, Burkitt lymphoma, and nasopharyngeal cancer increased by 29%, and their mortality increased by 17% [36]. Therefore, a therapeutic vaccine is urgently needed in order to prevent further increases in mortality of patients with EBV-related malignancies, but also for acute infections such as infectious mononucleosis development, which can also cause severe symptoms such as long-lasting lymphadenopathy and fever.

### 2.1. Prophylactic Vaccines

The main strategy of the prophylactic EBV vaccine is to develop neutralizing antibodies that would inhibit the EBV infection of host cells, mainly being B lymphocytes and epithelial cells [3]. There are different types of potential prophylactic vaccines and include vaccines based on proteins, antibodies, and epitopes (Figure 2). In addition, recombinant vaccinia virus, as well as recombinant adenoviral vectors, have often been used as vectors for gene entry. The main target is gp350/220 glycoprotein, which is the most abundant protein on the EBV surface. Gp350 vaccine can be used with gp350 subunit vaccine as an antigen, which triggers the attack of neutralizing antibodies and prevents infection. On the other hand, CD8+ T lymphocyte epitope vaccines based on gp350/220 protein trigger T lymphocytes aim to recognize EBV-infected cells that release cytotoxic granules and, consequently, trigger apoptosis. Vaccines based on epitopes prevent the further spread of infection and incidence of post-transplant lymphoproliferative diseases rather than preventing the primary infection [37]. The main domain needed for the attachment to CR2 on B lymphocytes is the N-terminal region of gp350/220 with amino acids 1–470, hence certain vaccines contain only fragments of the gp350/220 protein that possesses this region. Such protein subunits were expressed in clinical trials using Chinese ovary cells (CHO) for developing the gp350 vaccine, which successfully induced neutralizing antibodies in rabbits. Additionally, this trial showed that the soluble gp350 monomer triggers immunogenicity, with the same result obtained with and without adjuvant addition. Phase II showed a lower incidence of infectious mononucleosis in the vaccinated group compared with the placebo group; however, the vaccine did not manage to prevent initial EBV infection [19]. The gp350 vaccine with aluminum hydroxide as an adjuvant was tested in EBV-seronegative patients in order to prevent incidence of lymphoproliferative diseases by inducing gp350 neutralizing antibodies. However, an immune response against gp350 was not achieved in most children, which indicated the need for stronger adjuvants [38]. Other glycoproteins that participate in fusion such as gH, gL, and gB were prophylactic vaccine targets as well. Antibodies from rabbits previously immunized with gH/gL or trimeric gB and injected into humanized mice decreased EBV viremia and protected them from death after a lethal EBV dose [3]. Although the licensed prophylactic vaccine does not seem likely to be in use in the near future, research from early 2022 indicates promising results. The bivalent EBV vaccine made from two target antigens, gp350 and gH/gL/gp42 single chain, formed in nanoparticles as a result of bacterial ferritin, prevents both EBV infection and incidence of EBV-related lymphoma using humanized mice that had passive transfer of IgG [39].

### 2.2. Therapeutic Vacciness

Therapeutic vaccines are predestined for use after primary infection has already occurred, in order to alleviate or inhibit further spread or development of an acute infection as infectious mononucleosis. Therefore, therapeutic vaccines produced for EBV infection are targeted at patients with susceptibility for EBV-related malignancies or posttransplant lymphoproliferative diseases. Most of the clinical trials for therapeutic vaccines have been performed on patients with nasopharyngeal carcinoma [3]. EBV latency genes are the main targets of therapeutic vaccines, as well as genes responsible for lytic phase reactivation such as BZLF1 gene [40]. Target antigens for therapeutic vaccines come in contrast with prophylactic ones where envelope glycoproteins, required for entry in B lymphocytes, are used. Three strategies are most commonly used in EBV therapeutic vaccine development—dendritic cells, viral vectors, or a combination of both.

Dendritic cells are the most effective among antigen-presenting cells, with the capability of exogenous antigen presentation on MHC class I molecules as well as on MHC class II molecules, which makes them an effective vector for vaccine production [41]. Dendritic cells can be harvested from nasopharyngeal carcinoma patients and injected with the epitope of interest before being returned into the patients’ lymph nodes. A few latency genes, expressed in Hodgkin’s lymphoma and nasopharyngeal carcinoma, such as *EBNA1*, *LMP1*, and *LMP2*, represent potential epitopes for the EBV therapeutic vaccine, with which dendritic cells can be transformed [17]. Peptide epitopes used for therapeutic vaccines are mostly fragments of *EBNA1* and *LMP2* genes, which are responsible for cell transformation in nasopharyngeal carcinoma, certainly *EBNA1* and *LMP2* genes are expressed in latency II inside one hundred percent of EBV-related nasopharyngeal carcinoma. However, although dendritic cells can successfully activate both CD4+ and CD8+ T lymphocytes, the balance between cells that target EBNA1 and LMP2 epitopes, which are CD4+ T lymphocytes and CD8+ T lymphocytes, respectively, still has to be achieved. The response of both CD4+ and CD8+ T lymphocytes was detected in research using the chimeric antigen construct designed using the full-length *LMP2* gene and fragments of the C-terminal domain of *EBNA1* gene rich in CD4+ epitopes [42].

In therapeutic vaccines that contain viral vectors, the vaccinia virus Ankara modified with the genes of interest, such as *EBNA1* or *LMP1*, is the one most commonly used. Modified vaccinia virus Ankara is a replication-deficient viral vector firstly developed in chicken fibroblasts for vaccine production during smallpox eradication. It enters target cells and, owing to their modification with the gene of interest, activates their machinery for protein production [43]. The use of viral vectors has the additional advantage of enhancing the immune response through the expression of pathogen-associated molecular patterns [19]. The second most widely used viral vector is adenovirus, because it is able to induce an antigen-specific T lymphocyte response. The main disadvantage of using adenoviruses is pre-existing immunity in the host, which can neutralize it. Chimpanzee adenoviruses show a good response only after an initial dose, hence a second dose is usually provided with Ankara as a vector. Although humans do not have pre-existing immunity for chimpanzee adenoviruses, neutralizing antibodies are quickly developed after the initial dose [44].

The combination of dendritic cells and viral vectors is often used to stimulate an immune response by targeting B lymphocytes with latent genes. For example, autologous dendritic cell transduced with recombinant serotype 5 adenovirus encoding *LMP1* and *LMP2* was used to stimulate T lymphocytes; however, a response against *LMP1* or *LMP2* was not achieved [17]. Combinations of vaccine strategies have also been used with protein-based vaccines and viral vectors. Protein-based vaccines such as EBNA1 or LMP2 are also used as therapeutic vaccines. However, the immune response achieved by this vaccine strategy is usually based only CD4+ T lymphocytes. As the use of viral vectors is primarily based on CD8+ T lymphocyte response induction, the combination of viral vectors and protein-based vaccines is often used to achieve complete the immune response needed to eliminate the spread and progression of infection [44].

## 3. Ongoing Clinical Trials and Future Research

According to the clinical trials register “ClinicalTrials.gov”, there are currently two recruiting clinical trials for the EBV vaccine—one based on mRNA-1189 and one based on gp350–ferritin nanoparticle. Moderna started phase I clinical trials on 28 December 2021 with the EBV vaccine based on mRNA-1189, which contains five different mRNA coding for gp350, gB, gH, gL, and gp42 (Figure 3), in order to prevent primary EBV infection and incidence of infectious mononucleosis (https://www.sec.gov/Archives/edgar/data/1682852/000168285221000006/mrna-20201231.htm, accessed on 3 July 2022). The trial is randomized and placebo-controlled among 18–30-year-old healthy adults with an estimated completion date of 24 June 2023 (https://www.clinicaltrials.gov/ct2/show/NCT05164094?cond=EBV+vaccine&draw=2&rank=2, accessed on 29 June 2022). This is the first trial based on mRNA used in EBV vaccine development, encouraged by the mRNA vaccine for SARS-CoV2. It began after the previous demonstration of the mRNA vaccine candidate coding gp350, gp42, gH, gB, and gL, which induced a higher concentration of neutralizing antibodies in mice [3]. On the other hand, the National Institute of Allergy and Infectious Diseases started a phase I study on adjuvanted gp-350 ferritin vaccine on 27 November 2020. The trial is still in the recruiting phase with criteria of 20 EBV-seropositive and EBV-seronegative healthy adults aged 18–29. The start date of this trial is 29 March 2022, with a completion date of 1 July 2023 (https://www.clinicaltrials.gov/ct2/show/NCT04645147?cond=EBV+vaccine&draw=2&rank=3, accessed on 29 June 2022). The gp350–ferritin nanoparticle vaccine uses ferritin protein as a nanoparticle that provides a surface for antigen presentation. Ferritin is self-assembling intracellular protein that has been used as a platform for antigens among many vaccines including those for SARS-CoV2, influenza, and HIV-1. Part of gp350 used in this clinical trial is the N-terminal region, with amino acids 1–470 being the most important domain for gp350 attachment to CR2 [45].

Besides vaccine development, mostly in order to prevent PTLD, there are other development strategies with the aim to prevent EBV-related malignancies. Those make use of adoptive T lymphocyte therapy, EBV-specific T lymphocyte therapy, as well as EBV-specific chimeric antigen receptor T lymphocyte therapy. For instance, the end of the recent phase III clinical trial on cell therapy in lymphoproliferative diseases showed that more than 80% of PTLD patients remained alive after the initial treatment with allogeneic (Atara Biogenetics) and EBV-specific T lymphocyte therapy (https://www.healio.com/news/hematology-oncology/20220429/cell-therapy-a-meaningful-advance-for-posttransplant-lymphoproliferative-disease, accessed on 3 July 2022).

## 4. Conclusions

Although research into EBV vaccine development, both prophylactic and therapeutic, has been ongoing for a few decades, a licensed vaccine will not be available soon. So far, progress made in prophylactic and therapeutic EBV vaccine development suggests a conclusion that the first licensed EBV vaccine will be a therapeutic one rather than a prophylactic one. To entirely eliminate primary EBV infection with a prophylactic vaccine seems unfeasible and too much of a challenge for the near future before further understanding of its complex infection mechanism as well as a wide range of tropism for host cells. More fundamental research about the mechanism of entry into T lymphocytes as well as epithelial and NK cells needs to be conducted in order to achieve immunity in all EBV targets. Promising results in inducing both CD4+ and CD8+ T lymphocyte response showed dendritic cells, although the number of antigen epitopes they can present is limited and their production is not cost-effective, while the usage of Chinese hamster ovary cells showed desired induction of neutralizing antibodies with no specific adjuvant needed However, all positive results are in favor of the further spread of infection and inhibition of the incidence of lymphomas rather than inhibition of primary infection. In fact, most phase II clinical trials showed a lower incidence of infectious mononucleosis, but still did not manage to inhibit initial attachment of gp350 to CR2 because of the presence of infection. Gp350 is the most abundant glycoprotein on EBVs’ envelope; therefore, the presence of EBV acquires a high number of gp350-epitope specific T-lymphocytes to maintain further development of infection. Still, a strong T lymphocyte response will not be enough to eliminate infection, but all neutralizing antibodies and the CD4+ and CD8+ T lymphocyte respond together. Up to now, vaccines have shown an impact in only one area of immune response, more or less effectively. Accordingly, the combination of vaccine strategies is recent and the future course for vaccine development involves two different vectors that induce different immunological responses used in different doses. Recruiting clinical trials involve new strategies using groups of targeted mRNAs inside nanoparticles. mRNA-based vaccines were not used until now in EBV vaccines, hence future results could indicate the best material for its development. The usage of multiple mRNAs or peptide epitopes inside nanoparticles indicated the best chance for a licensed vaccine. The vaccine needs to include as a wide range of EBV target cells as possible in order to eliminate the development of infection as much as possible. However, in order to know which target antigens to incorporate into vectors, a deeper understanding about the EBV replication cycle, entry into target cells, and its interaction with immune system needs to be achieved.

## Figures and Tables

**Figure 1 pathogens-11-00864-f001:**
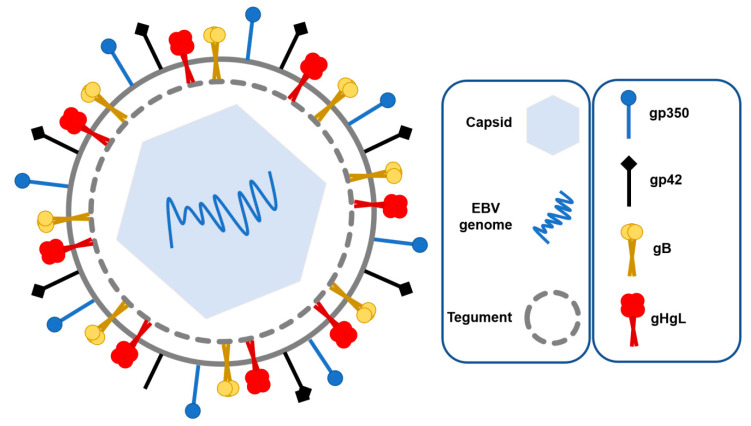
Structure of EBV virion; stressed glycoproteins gp350, gHgL, gp42, and gB are required for attachment and fusion with B lymphocytes during primary infection.

**Figure 2 pathogens-11-00864-f002:**
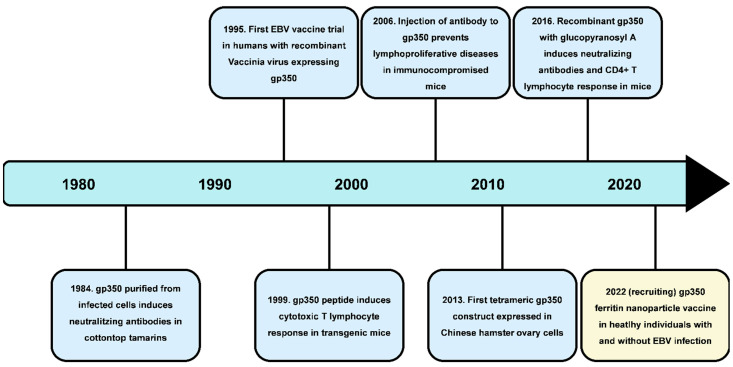
Development of the production strategy of the EBV prophylactic vaccine based on gp350 protein.

**Figure 3 pathogens-11-00864-f003:**
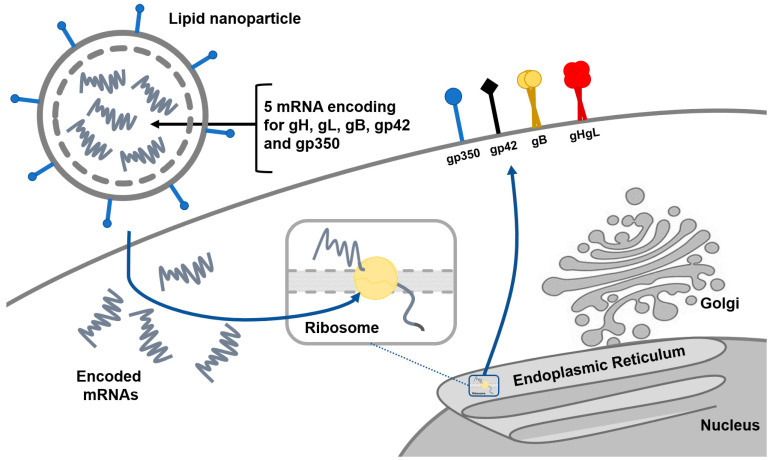
EBV vaccine designed with mRNA-1189, with a group of five different mRNAs coding for glycoproteins needed for virus attachment and fusion, incorporated into lipid nanoparticles. Translated proteins are displayed on the surface of B lymphocytes, which induces the production of neutralizing antibodies.

**Table 1 pathogens-11-00864-t001:** Genes expressed in different EBV latency types. Latency type 0 has no genes expressed, but small non-coding RNAa EBER localized in the nucleus. The most abundant gene expression profile is present in latency type III present in EBV transformed B lymphocytes.

Latency Type	Expressed Genes	Cells
**0**	EBERs (noncoding RNAs)	Memory B lymphocytes
**I**	EBERs, EBNA1	Memory B lymphocytes
**II**	EBERs, EBNA1, LMP1, LMP2A, LMP2B	Proliferating and maturing EBV-infected lymphocytes
**III**	EBERs, EBNA1, EBNA2, EBNA3A, EBNA3B, EBNA3C LMP1, LMP2A, LMP2B	Naive B lymphocytes

## Data Availability

Not applicable.

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
