# Peer review of "Progress in Prophylactic and Therapeutic EBV Vaccine Development Based on Molecular Characteristics of EBV Target Antigens"

_pathogens, 2022, doi:10.3390/pathogens11080864_

Round 1
Reviewer 1 Report
his manuscript reads well with minor grammatical and/or typographical errors, and provides available information on the progress of EBV vaccine research.
Comments:
-The authors need to provide more references for the sentence/information in Introduction section.
-There have been instances that the authors have depended on one publication for a host of information which they could have gotten from different sources, such as 1.3. Interaction of EBV and immune system, only 2 literatures are cited. It is not enough for the information in one review to come from 32 references.
-Page2 line61-62 “Precise mechanism of EBV entry into epithelial cells is still not know”. Regarding the mechanism of EBV entry into epithelial cells, there are some studies in recent years, please supplement and add references.
-As a virus with an infection rate of more than 95%, the authors should discuss more about the need for vaccine research, especially for therapeutic vaccines.
Reviewer 2 Report
The manuscript is fairly well written, but requires some revisions to improve presentation quality and refinement as follows.
Please include some addition very recent references (2022) in the Introduction to make the paper most current.
The resolution of Figure 2 is low (the letters are somewhat fuzzy). Increase the resolution to at least 300 dpi (or higher) for greater clarity. Please also change the background of the blue panels to lighter blue and change the text to bold black to increase contrast for all panels.
There are a number of misspelled words in the manuscript. Please do a spell check on the manuscript. I am mentioning just a few misspelled words I found as follows, but there are probably more. The proper spelling of these words are given below.
L 189 vaccines, plus the entire sections 2.1 and 2.2 (have misspelled this word)
L 326 unfeasible
L 343 Accordingly,
Round 2
Reviewer 1 Report
The authors have adequately addressed my initial concerns and accordingly revised their manuscript. I have no further questions, comments or suggestions.